

# Efficient Bootstrap Estimates for Tail Statistics

Øyvind Breivik[1,2] and Ole Johan Aarnes[1]

[1]Norwegian Meteorological Institute
[2]Geophysical Institute, University of Bergen

*Correspondence to:* Øyvind Breivik, Norwegian Meteorological Institute, Allegaten 70, NO-5007 Bergen, Norway. E-mail:
oyvind.breivik@met.no. ORCID Author ID: 0000-0002-2900-8458

**Abstract.** Bootstrap resamples can be used to investigate the tail of empirical distributions as well as return value estimates based on the extremal behaviour of the distribution. Specifically, the confidence intervals on return value estimates or bounds on in-sample tail statistics can be estimated using bootstrap techniques. However, bootstrapping from the entire data set is expensive. It is shown here that it suffices to bootstrap from a small subset consisting of the highest entries in the sequence to make estimates that are essentially identical to bootstraps from the entire sequence. Similarly, bootstrap estimates of confidence intervals of threshold return estimates are found to be well approximated by using a subset consisting of the highest entries. This has practical consequences in fields such as meteorology, oceanography and hydrology where return estimates are routinely made from very large gridded model integrations spanning decades at high temporal resolution. In such cases the computational savings are substantial.

## 1   Introduction

Bootstrap resamples of time series are commonly used to estimate confidence intervals on return values (Naess and Clausen, 2001; Naess and Hungnes, 2002) and to investigate the behaviour of the tail of the empirical distribution (Coles, 2001; Beirlant et al., 2006; Qi, 2008). This is a straightforward procedure, but one which quickly becomes cumbersome for large data sets as it demands random draws from the entire data set which subsequently must be sorted to get to the upper percentiles. When handling long model integrations in meteorology and oceanography with spatially gridded fields of typically $10^6$ grid points this brute-force approach becomes impractical. Such quantities are regularly encountered when estimating return levels from atmospheric reanalyses (Kalnay et al., 1996; Saha et al., 2010; Compo et al., 2011; Dee et al., 2011; Poli et al., 2016), wave hindcasts (Swail and Cox, 2000; Caires and Sterl, 2005; Gaslikova and Weisse, 2006; Breivik et al., 2009; Reistad et al., 2011; Aarnes et al., 2012) and long climate integrations that cover decades or even centuries (Hersbach et al., 2015). When even larger data sets are used, such as the ensembles of seasonal integrations (Stockdale et al., 2011; Molteni et al., 2011), as was done by Van den Brink et al. (2005) on a data set amounting to nearly 1,000 years, the data processing becomes nearly intractable for spatially extensive fields and findings ways to reduce the size of the data sets becomes essential. That is the subject of this paper.

We will present a simple argument for why it is sufficient to retain only a small subset $K_0$ consisting of the highest entries in a data set when estimating tail statistics such as return levels and their associated confidence intervals by means of bootstrapping.



These highest entries will normally only represent a small fraction of the total data set. This reduces the need for sorting and storage by several orders of magnitude. The method also reduces the task of sorting the original data set as only the $K_0$ highest entries are kept.

This paper is organised as follows. Sec 2 presents the binomial argument for why we can bootstrap from a small subset consisting of the highest entries in the original data set. Sec 3 presents three examples of bootstrapped confidence intervals of various tail statistics for a data set of significant wave height from the central North Sea. Here we also show how the method laid out in Sec 2 can be used in practice to determine how many entries must be kept in order to perform an unbiased bootstrap. Sec 4 summarises the results and presents the conclusions.

## 2 Bootstrapping from the $K_0$ highest entries in a data set

Consider the sequence $\mathcal{D}_0$ of independent and identically distributed (iid) random numbers $X_1, X_2, \ldots, X_N$. Let $X_{N,1} \leq X_{N,2} \leq \cdots \leq X_{N,N}$ denote the order statistics on $\mathcal{D}_0$. When investigating a statistic $\theta$ which is a function of the $k$ highest entries in $\mathcal{D}_0$, ie $\theta = f(X_{N,N-k+1}, X_{N,N-k+2}, \ldots, X_{N,N})$, it is common to form $M$ bootstrap resamples $\mathcal{D}_1, \mathcal{D}_2, \ldots, \mathcal{D}_M$, each of length $N$ (Diaconis and Efron, 1983; Efron and Gong, 1983). This method can be used to compute the confidence intervals around extreme value estimates (Breivik et al., 2013, 2014). The procedure is computationally intensive and memory-consuming, as it involves bootstrapping and storing $M \times N$ numbers and performing $M$ sorts, each a process of $\mathcal{O}(N \log N)$ operations. Since we are only interested in combinations of the $k$ highest entries in the resamples $\mathcal{D}_1, \mathcal{D}_2, \ldots, \mathcal{D}_M$, we will explore the possibility of instead resampling from only the highest $X_{N,N-K_0+1}, X_{N,N-K_0+2}, \ldots, X_{N,N-k+1}, \ldots, X_{N,N}$ entries in $\mathcal{D}_0$ $(K_0 > k)$. This will be referred to as the *resample threshold* and is sometimes more conveniently written as the percentage of data left out, $P_0 = 100(1 - K_0/N)$.

The probability of drawing one of the highest $K_0$ entries in $\mathcal{D}_0$ is a binomial problem with probability $p = K_0/N$. The probability of making exactly $k$ draws (with replacement) from the highest $K_0$ in $N$ draws is thus given by the binomial probability mass function [Zwillinger 1996, p 581]

$$f_{\mathrm{binom}}(k; N, p) = \mathrm{P}(X = k) = \binom{N}{k} p^k (1-p)^{N-k}. \tag{1}$$

where $X$ is a random variable representing the number of draws. The probability of drawing fewer than the required $k$ entries from the highest $K_0$ is given by the binomial cumulative distribution function

$$F_{\mathrm{binom}}(k-1; N, p) = \mathrm{P}(X < k) = \sum_{i=0}^{k-1} \binom{N}{i} p^i (1-p)^{N-i}. \tag{2}$$

A full bootstrap resample $\mathcal{D}_i$ of length $N$ from $\mathcal{D}_0$ will contain $K_i$ entries from the highest $K_0$, and $K_i \sim \mathrm{Binom}(N, p)$ where $E[K_i] = K_0$ since the expected value of the binomial distribution (1) is

$$\mu_{\mathrm{binom}} = Np = K_0. \tag{3}$$





The variance is

$$\sigma_{\text{binom}}^2 = Np(1-p) = K_0 - K_0^2/N \approx K_0 \text{ when } K_0 \ll N. \tag{4}$$

Denote a short bootstrap resample from the $K_0$ highest entries in $\mathcal{D}_0$ as $\tilde{\mathcal{D}}_i$. Two conditions must be met for $\tilde{\mathcal{D}}_i$ to be an unbiased substitute for $\mathcal{D}_i$:

1. The number $K_0$ must be set large enough that the probability that we miss entries smaller than $X_{N,N-K_0+1}$ in $\mathcal{D}_0$ is below a chosen threshold $p_c$.

2. The length $\tilde{K}_i$ of $\tilde{\mathcal{D}}_i$ must have the same mean and variance as $K_i$ (Eqs 3–4).

To fulfil Condition (1) it is sufficient to decide on an acceptable level for $p_c$. This probability can be found by consulting Eq (2). It is important to note that choosing $K_0$ too small will bias the statistic $\tilde{\theta} = f(\tilde{\mathcal{D}}_i)$ since it will be estimated from bootstrap samples that miss entries smaller than $X_{N,N-K_0+1}$. We will for this reason refer to $p_c$ as the *probability of contamination* as it gives the probability that the bootstrap estimate is biased because we have kept too few entries from the original data set $\mathcal{D}_0$. A very conservative bound on $p$, and thus on $K_0 = Np$, can be found quickly by consulting Hoeffding's formula (Hoeffding, 1963),

$$F(k; N, p) \leq \exp\left(-2\frac{(Np-k)^2}{N}\right), \tag{5}$$

valid when $k \leq Np$. A useful quantity is the ratio $r = K_0/k$ of upper entries retained ($K_0$) and the minimum number $k$ required to form a bootstrap estimate of the statistic in question for a given probability of contamination $p_c$. This can be estimated from Eq (2), but when $N$ is large the Poisson distribution is a good approximation and more practical to work with,

$$F_{\text{Poisson}}(k-1; rk) = \text{P}(X < k) = e^{-rk}\sum_{i=0}^{k-1}\frac{(rk)^i}{i!}. \tag{6}$$

Fig 1 shows the minimum acceptable ratio $K_0/k$ as a function of $k$ for levels of $p_c$ ranging from $10^{-5}$ to 0.05. The probabilities can be computed from Eq (2) [or more conveniently from Eq (6)]. As can be seen, for all values of $k$, the ratio is comfortably below 15, and for values of $k$ larger than 10 a ratio of 3 is sufficient even for a confidence level of $10^{-5}$. See the appendix for a more detailed explanation of the ratio curves used throughout.

Condition (2) can be handled by randomly perturbing the size of the resamples, $\tilde{K}_i$, such that it mimics the number of draws, $K_i \sim \text{Binom}(N, p)$, that would have been made from the upper $K_0$ entries of $\mathcal{D}_0$ in a full bootstrap $\mathcal{D}_i$. In practice, as we shall see, the statistics are quite insensitive to these perturbations as long as $K_0$ has been chosen sufficiently large.

## 3 Bootstrapping confidence intervals

Here we present worked examples of how the two conditions presented above can reduce the problem of estimating confidence intervals on tail statistics for a data set of independent ensemble forecasts at long lead time ($N = 330{,}000$). We use archived



ensemble forecasts (Molteni et al., 1996) of significant wave height in the central North Sea (near the Ekofisk oil field at $56.5°$ N, $003.2°$ E; a histogram of the data set used is shown in Fig 2) at a forecast lead time of 240 hours. 100-year return values from these ensembles have previously been reported by Breivik et al. (2013) and Breivik et al. (2014).

### 3.1 Example 1: Confidence intervals on in-sample return estimates

Consider as an example the problem of how to calculate in-sample return estimates from the data set of independent forecasts presented above. These forecasts can be considered iid (as they are not from correlated time series). An in-sample return estimate is calculated directly from the tail of the empirical distribution rather than by applying extreme value analysis. As explained by Breivik et al. (2013) the independent forecasts presented in Fig 2 add up to the equivalent of 229 years under the assumption that each forecast represents a time interval $\Delta t = 6$ hours. A 100-year return estimate is then a linear interpolation

between $X_{N,N-1}$ and $X_{N,N-2}$ (the second and third highest entries in $\mathcal{D}_0$),

$$H_{100} = 0.67X_{N,N-1} + 0.33X_{N,N-2}. \tag{7}$$

Now, clearly $k = 3$ since we need the second and third highest entries in our resamples to form a return estimate. Let us now tentatively keep the $K_0 = 1,000$ highest entries and bootstrap from these instead of from the entire sequence to compute the confidence intervals on the linear combination of the second and third highest entries given by Eq (7). The size $\tilde{K}_i$ of the

resamples, $\tilde{\mathcal{D}}_i$, is drawn from the binomial distribution (Eqs 3–4) with $\mu = K_0$ and $\sigma^2 \approx K_0$. What is the probability $p_c$ that one of the three highest entries in a bootstrapped sequence should *not* have come from the 1,000 highest entries that we have retained (i.e. should depend on entries contained in the bulk of the data set that we discarded)? It is clear that the probability of drawing one of the highest 1,000 entries is $p = 1,000/330,000$, and from Eq (2) we find that the probability of picking too few ($< 3$) entries from the $K_0$ highest is

$$F(2; 330,000, p) = \mathrm{P}(X \leq 2) = \sum_{i=0}^{2} \binom{330,000}{i} p^i (1-p)^{330,000-i}, \tag{8}$$

which is indistinguishable from zero to double precision. Reducing the number $K_0$ to 10 ($r \approx 3$) raises the probability of contaminating the resamples by entries from the lower $N - K_0$ to 0.002. This can also be confirmed by consulting Fig 1 for the combination $k = 3, r = 3$. For $M = 1,000$ resamples we may thus expect on average 2 resamples to be contaminated by values from the lower $N - K_0$ values in the original sequence. A very safe compromise in this case is $K_0 = 100$ ($r \approx 33$). Consulting

Fig 1 shows that for $k = 3, r = 33$ we are well below a probability of contamination of $10^{-5}$. The quantile-quantile (QQ) plot in Fig 3 shows that resampled return estimates of significant wave height from the full data set $\mathcal{D}_0$ (see Fig 2) have practically the same distribution as resamples from the upper $K_0 = 100$ entries.

Condition (2) given above states that the length of the reduced resamples $\tilde{\mathcal{D}}_1, \tilde{\mathcal{D}}_2, \ldots, \tilde{\mathcal{D}}_M$ should be randomly perturbed around the mean value $K_0$. In practice this condition turns out to be rather insignificant as long as $K_0$ is chosen sufficiently

large. This is demonstrated in the QQ plot in Fig 4 where we see that perturbed-length estimates (abscissa) closely match the distribution of fixed-length estimates (ordinate). However, choosing $K_0$ too small will bias the statistic in question. This





is illustrated in Fig 5 where we see that bootstrap estimates from too-short data sets ($K_0$ chosen too small) are biased high. As $K_0$ approaches 30 ($r = 10$), the mean and standard deviation of the return estimates approach their asymptotic values. These findings are in accordance with what we find by consulting Fig 1 where we see that $k = 3, r = 10$ has a probability of contamination $p_c$ less than $10^{-5}$.

## 3.2 Example 2: Confidence intervals on upper percentiles

A similar problem to the estimation of confidence intervals for in-sample return values is how to estimate the confidence interval for the highest percentiles, e.g. the 99th percentile ($P_{99}$). The upper percentile is frequently used when investigating trends in for example the wind and wave height climate [see e.g. Wang and Swail (2001, 2002)]. In order to construct a bootstrap estimate of $P_{99}$ brute force it is necessary to resample the entire data set $\mathcal{D}_0$ and sort the bootstrap to get to the $N/100$-th highest entry. However, Fig 1 tells us that when $k = N/100$ is large (as it will be when $N$ is large), we can with extremely high certainty say that keeping the $K_0 = 2k$ highest entries is enough to perform a bootstrap resample exercise for the confidence interval on $P_{99}$. In fact, $K_0 = 1.2k$ is sufficient for all significance levels plotted in Fig 1. This means that in order to estimate a confidence interval for $P_{99}$ we need only find the entry $X_{N,N-k}$ that corresponds to $P_{99}$ from the original data set $\mathcal{D}_0$ and retain entries higher than $X_{N,N-1.2k}$. Fig 6 shows how the ratio $r$ decreases as the sample size $N$ increases. It is clear that for all probabilities of contamination investigated, a ratio of $K_0/k = 2$ is sufficient when $N$ is larger than 2,000. Obviously, samples smaller than $\mathcal{O}(10^3)$ do not pose computationally demanding problems anyway and are of no interest to us in this context. Fig 7 illustrates for a fixed probability of contamination $p_c = 0.01$ that even as we go to higher percentiles (the uppermost curve shows $P_{99.9}$), a ratio $K_0/k = 2$ is sufficient as the sample size $N$ exceeds $10^4$ (see the appendix for more details on the ratio curves).

## 3.3 Example 3: Confidence intervals on return estimates from threshold exceedances

Consider now the problem of estimating confidence intervals for threshold exceedances. The Generalized Pareto distribution (GPD) gives the relevant extreme value distribution for independent exceedances above a threshold $u$ (Coles 2001, pp 75–77),

$$H(y) = 1 - \left(1 + \frac{\xi y}{\tilde{\sigma}}\right)^{-1/\xi}. \tag{9}$$

Here $y = X_i - u, y > 0$ are exceedances above an entry $X_k$, $\tilde{\sigma}$ is a scale parameter which is a function of the threshold $u$, and $\xi$ is the shape parameter.

A brute-force approach would be to make $N$ draws from $\mathcal{D}_0$ (with replacement) and repeat this procedure $M$ times. Then, GPD return estimates would be computed for each of the resulting bootstrap sequences $\mathcal{D}_1, \mathcal{D}_2, \ldots, \mathcal{D}_M$. Say we want to try to instead keep only the $K_0$ entries exceeding a threshold $U_0 < u$ corresponding to the entry $X_{K_0}$ in the original data set $\mathcal{D}_0$. From these we need to draw at least $k$ entries, from which we will make return estimates. The question is again how many entries ($K_0$) must be kept to arrive at an acceptably low probability $p_c$ that the statistic should really be based on entries below the threshold $U_0$.



This problem arises when estimating GPD return values from the independent ensemble forecasts (Fig 2). For such a data set all exceedances above a given threshold can be used to form GPD return value estimates (9). Confidence intervals on the return values can likewise be estimated by bootstrapping from all entries exceeding this threshold. For a large data set this is orders of magnitude faster than bootstrapping from the entire data set. Assume again that we have kept all forecasts exceeding

$P_{99.1}$, ie the $K_0 = 3,000$ highest entries (cf Fig 2). To form a return estimate we assume that we need at least $k = 1,000$ entries, corresponding to $P_{99.7}$. The probability of drawing (with replacement) $k$ or fewer entries from the highest $K_0$ in $N$ draws can again be found from Eq (2) and is indistinguishable from zero to double precision with the given choice of $N$, $K_0$ and $k$. This is easy to verify by consulting Fig 1 where we see that for $k = 1000, r = 3$ we are well above the $10^{-5}$ level. Fig 8 shows that the confidence interval and the mean return value based on $M = 1,000$ bootstrap resamples for various choices of resample

threshold $100(1 - K_0/N)$ (i.e. the percentage of data omitted) are practically identical to the confidence intervals based on the full data set $\mathcal{D}_0$ (marked as asterisks). Only when $r = K_0/k$ comes close to unity do we experience fluctuations and biases (i.e., the resample threshold nearly coincides with the number of tail entries required to form a return estimate, in this case the threshold $P_{99.7}$).

## 4   Conclusions

Confidence intervals and other statistics on the extremes and the tail of empirical distributions are commonly found using bootstrap techniques. Here we have shown that it is unnecessary to bootstrap from the entire data set. The actual number $K_0$ highest entries that must be kept to make unbiased bootstrap estimates for the tail of an empirical distribution depends on $K_0 = Np$ as well as on the number $k$ highest entries that are required for the statistic in question. The examples in the previous sections calculated $p_c$ given a predetermined number $K_0$ of tail entries that have been kept. This is a realistic approach as in

practice we often retain a larger part of the tail of an empirical distribution than what is strictly needed since the same data set is used to compute other statistics. It is then sufficient to consult Eq (2) to determine whether $K_0$ is sufficiently large. A quick estimate of the probability of contamination can be made by consulting Fig 1.

The advantages of restricting resamples to a small subset $K_0$ consisting of the highest entries in $\mathcal{D}_0$ can be summarised as follows. First, only the upper $K_0$ entries need be kept and sorted in the original data set. This offers substantial savings in

cases like those described by Breivik et al. (2013, 2014) where a very large number of forecasts ($> 300,000$) are handled, each consisting of more than 60,000 grid points in space. Second, the size of the resamples is also reduced from $N$ to an average size $K_0$, where $K_0$ is usually a very small fraction of $N$, typically less than 1%. Third, this reduction in resample size also means that the cost of sorting the resamples to get to the highest entries is greatly reduced.

## Appendix:   Consulting the ratio curves

The ratio curves presented in Figs 1, 6 and 7 are convenient for quickly establishing how many entries ($K_0$) must be kept in order to form an unbiased resample that depends on the highest $k$ entries. The relationship between Fig 1 and Fig 6 can be





illustrated as follows. If we assume $N$ large we can use Fig 1. In practice we can choose $N = 2 \times 10^3$ without violating the assumption that $N$ is large. Now assume that the statistic in question is the 99th percentile, i.e. $k = N/100 = 20$. Let us choose a probability of contamination $p_c = 0.01$ (this corresponds to the red curve marked with diamonds in Fig 1). We find the ratio to be 1.6, i.e. we will need to keep 60% more entries than the entry corresponding to $P_{99}$. The corresponding curve in Fig 6 is

5    also marked in red. Here, the location on the $x$-axis to read off is $N = 2 \times 10^3$ which lies on the $y$-axis, and the ratio is again found to be 1.6. A more realistic example in terms of sample size would be $N = 10^5$ (and $k = N/100 = 10^3$). Now we find from either Fig 1 or Fig 6 that with a probability of contamination $p_c = 0.01$ that the ratio is 1.13, i.e. we need only keep 13% more entries than the one representing the 99 percentile. Figs 1, 6 and 7 clearly illustrate that in almost all cases it is sufficient to retain at most twice as many entries $K_0$ from the tail of the sample distribution $\mathcal{D}_0$ than what is required ($k$) for the statistic

10    in question.

*Acknowledgements.* This study was carried out with support from the Research Council of Norway through the ExWaCli project (grant no 226239). The data sets presented in this study are archived in the MARS database of the European Centre for Medium-Range Weather Forecasts (ECMWF), see http://old.ecmwf.int/services/archive/.



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

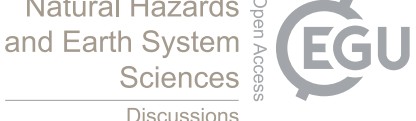



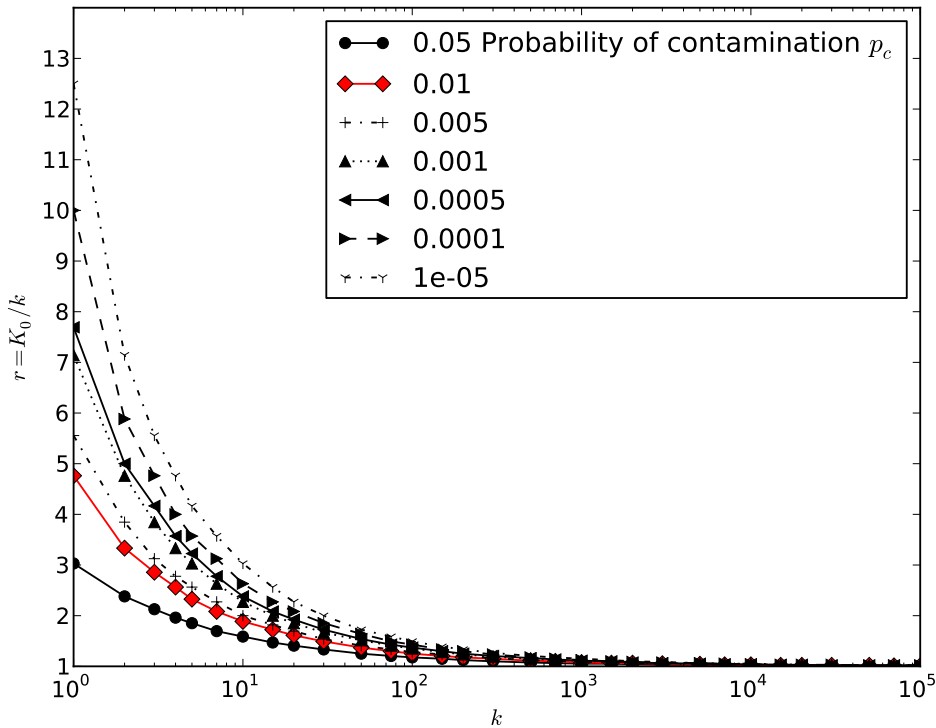

**Figure 1.** The ratio $K_0/k$ as a function of $k$, the minimum number of bootstrapped entries needed for the statistic in question, for levels of probability of contamination ranging from $10^{-5}$ (uppermost curve) to $0.05$ (lowermost curve). The curve representing 1% probability of contamination is marked in red (with diamonds) as it is a reasonable confidence level.



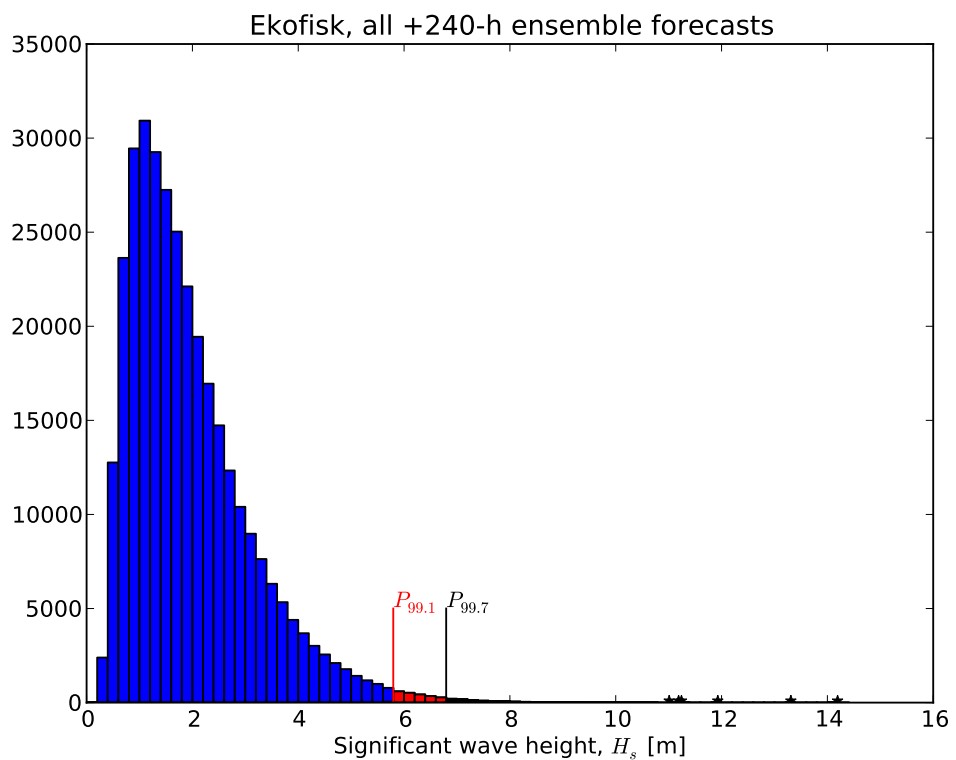

**Figure 2.** Histogram of the significant wave height from archived ensemble forecasts in the central North Sea (Ekofisk, 56.5N, 003.2E) at +240 h lead time. Entries above $P_{99.1}$, corresponding to threshold $U_0$, are coloured red whilst entries exceeding $P_{99.7}$, corresponding to the upper threshold, $u$, are in black. The highest entries are individually marked with asterisks.





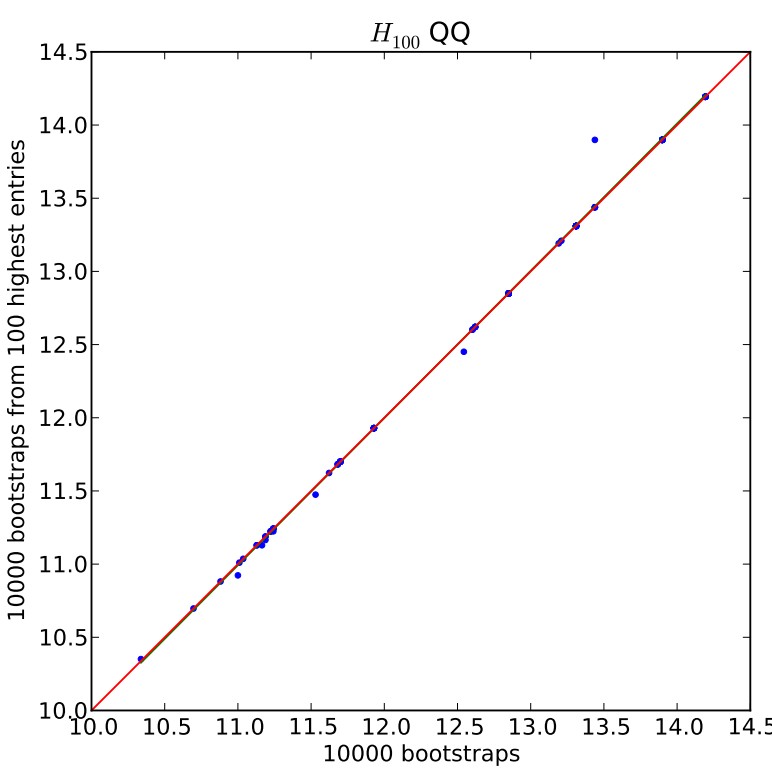

**Figure 3.** A quantile-quantile comparison of 10,000 bootstrapped direct 100-year return estimates of significant wave height taken from a forecast ensemble (Breivik et al., 2013) versus a bootstrap from the upper 100 entries in the data set. The $45°$ line is shown in red.





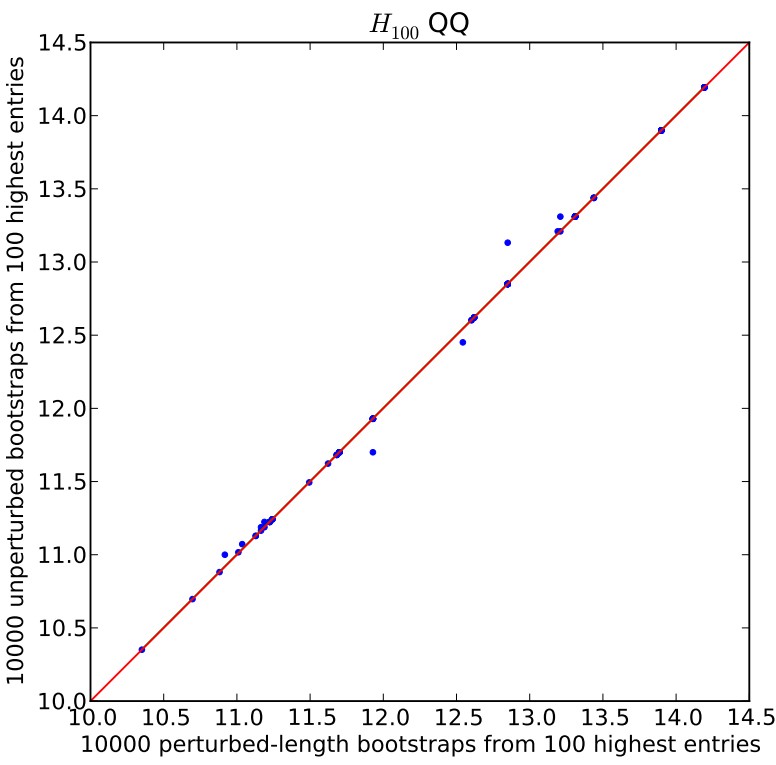

**Figure 4.** A quantile-quantile (QQ) comparison of $M = 10,000$ bootstraps $\tilde{\mathcal{D}}_1, \tilde{\mathcal{D}}_2, \ldots, \tilde{\mathcal{D}}_M$ of variable length $\tilde{K}_1, \tilde{K}_2, \ldots, \tilde{K}_M$ against bootstraps of fixed length $K_0$, all from the upper 100 entries in the original sequence $\mathcal{D}_0$. The difference is very small.





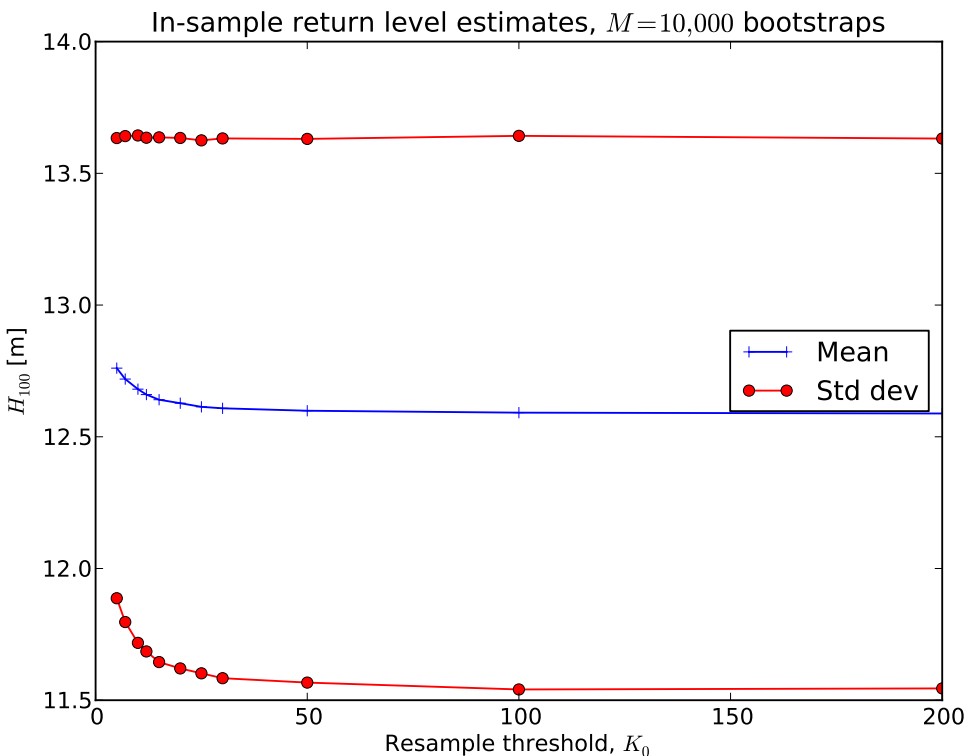

**Figure 5.** Mean and standard deviation on 100-yr in-sample return estimates based on $M = 10,000$ bootstrap resamples for various choices of resample threshold $K_0$ for the data set in Fig 2. A minimum of $k = 3$ entries are required to form the return estimate [see Eq (7)]. For choices of $K_0$ smaller than 30 (corresponding to a ratio $r = K_0/k = 10$) the bootstrap resamples are biased high.





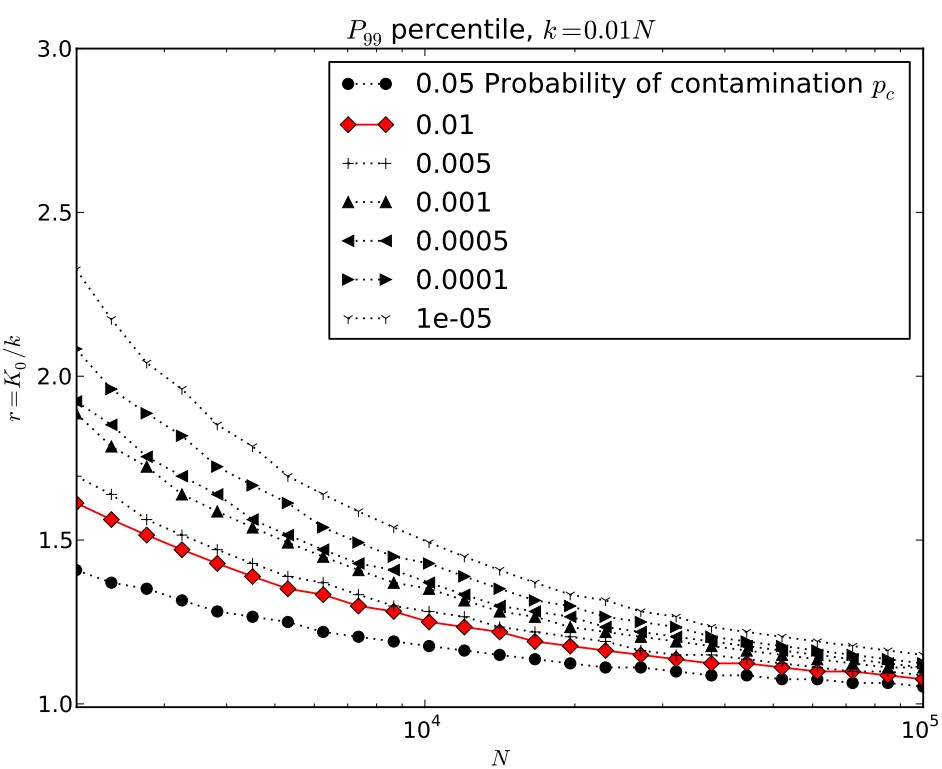

**Figure 6.** Bootstrapping the 99th percentile, $P_{99}$. The ratio $r = K_0/k$ is shown as a function of sample size $N$. Here, the minimum number of entries required is simply the upper 1% ($P_{99}$), so $k = N/100$. Various levels of probability of contamination $p_c$ are shown, and for sample sizes larger than approximately $2,000$, a ratio $r = 2$ is sufficient. The curve representing 1% probability of contamination is marked in red (with diamonds) as it represents a reasonable confidence level.





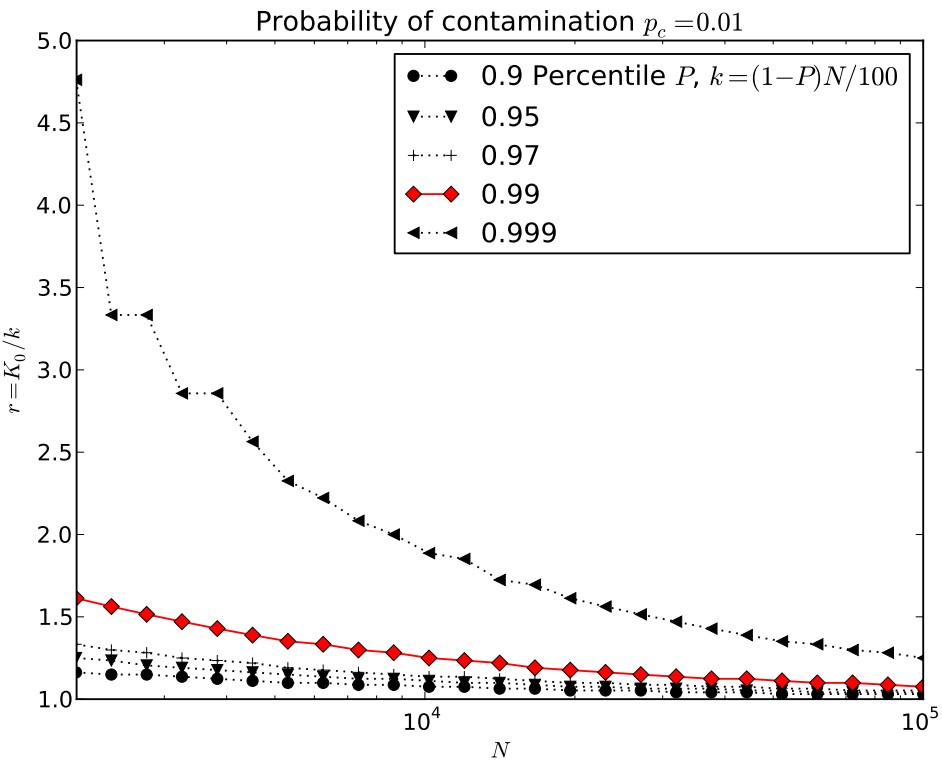

**Figure 7.** Bootstrapping the upper percentiles $P = P_{90}, P_{95}, P_{97}, P_{99}$ and $P_{99.9}$. The ratio $r = K_0/k$ is shown as a function of sample size $N$. Here, the minimum number of entries required is $k = (1-P)N/100$. The probability of contamination is kept fixed at $p_c = 0.01$. At sample sizes larger than approximately $10^4$, a ratio $r = 2$ is sufficient for all percentiles investigated. The curve representing the 99th percentile is marked in red (with diamonds) and corresponds to the red curve in Fig 6.





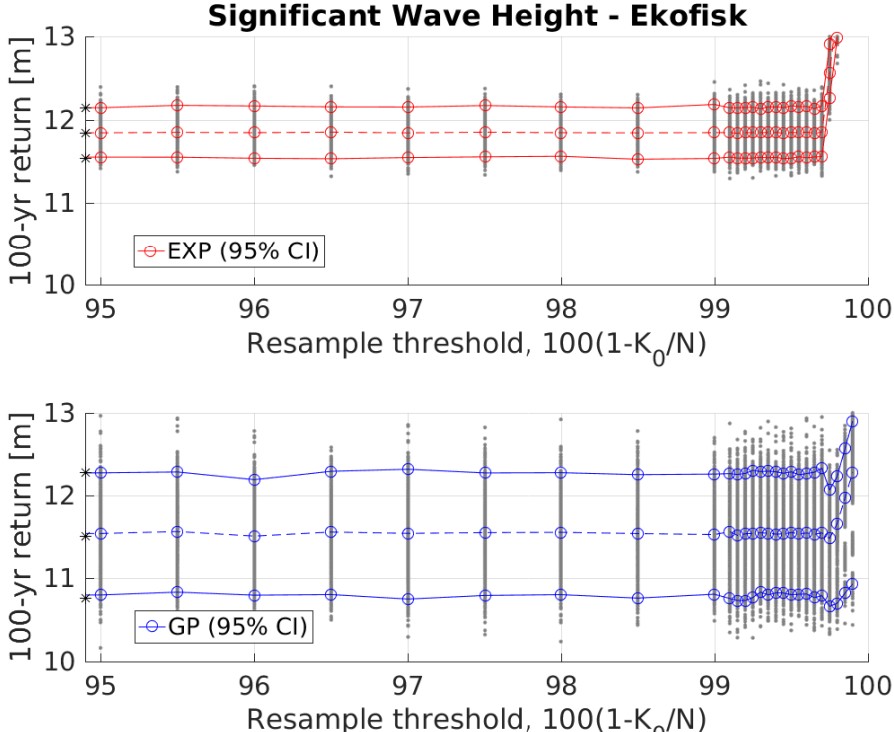

**Figure 8.** The upper and lower 95% confidence intervals and the mean 100-yr return estimates (dashed) based on $M = 1000$ bootstrap resamples for various choices of resample threshold $K_0$ for the data set in Fig 2. Upper panel: a GPD with shape parameter $\xi = 0$ (exponential distribution). Lower panel: a GPD with freely varying shape parameter. Individual bootstrap estimates are marked in grey. Estimates based on the full data set $\mathcal{D}_0$ are marked as asterisks on the ordinate.