# Peer review of "Efficient Bootstrap Estimates for Tail Statistics"

_Natural Hazards and Earth System Sciences, 2016_

## Referee Comment (RC1) · R. Katz (Referee) · 23 Nov 2016

How to Bootstrap Extremes if You Must

GENERAL COMMENTS:

The focus of the manuscript is on efficient use of the bootstrap, a resampling technique, to quantify uncertainty (e.g., in the form of a confidence interval) in estimated extreme statistics such as return levels. Justification is provided for a simplified bootstrap procedure in which the resamples are generated through only drawing from the highest values in the original sample, not the entire sample. This common sense result is consistent with conventional statistical modeling of extremes, with the common assumption that the uncertainty in estimating the rate of exceedance of a high threshold can be ignored (e.g., Chapter 4 in Coles, 2001). Perhaps the present paper serves to

place this conventional approach on firmer footing.

Nevertheless, there are a number of alternative techniques for uncertainty quantification in extreme value analysis not even mentioned in the manuscript. These alternatives include different implementations of the bootstrap, as well as ones in which no resampling need be performed (e.g., profile likelihood technique; Coles, 2001). At the least, these alternatives should be mentioned.

For this reason, I recommend that the manuscript be accepted for publication subject to minor revision.

SPECIFIC COMMENTS:

(1) Nonparametric versus parametric bootstrap

A nonparametric bootstrap is used in which the resamples are created by drawing with replacement from the original sample. When fitting extreme value distributions (e.g., the generalized Pareto in Sec. 3.3), it has been suggested that a parametric bootstrap would be preferable for constructing confidence intervals for return levels (i.e., resamples are created by Monte Carlo simulation from the fitted distribution) (Kysely, 2008).

(2) Refined bootstrap techniques

Bootstrap-based confidence intervals can be too short, especially for return levels with long return periods. Consequently, alternative more involved bootstrap techniques (e.g., the so-called "test inversion" bootstrap) have been proposed to improve the performance of such confidence intervals (Schendel and Thongwichian, 2015).

(3) Alternatives to bootstrap

When estimating the parameters of an extreme value distribution by maximum likelihood, an alternative technique for obtaining confidence intervals for return levels is profile likelihood (Coles, 2001). This technique does not require any resampling, but does
require repeated fits of the extreme value distribution under parameter constraints. It is competitive with resampling for obtaining confidence intervals of return levels (e.g., Schendel and Thongwichian, 2015).

REFERENCES:

Kysely, J.: A cautionary note on the use of nonparametric bootstrap for estimating uncertainties in extreme-value models, Journal of Applied Meteorology and Climatology, 47, 3236–3251, 2008.

Schendel, T. and Thongwichian, R.: Flood frequency analysis: Confidence interval estimation by test inversion boostrapping, Advances in Water Resources, 83, 1–9, 2015.
* * *

---

## Referee Comment (RC2) · S. Caires (Referee) · 29 Dec 2016

General comments:

The authors show that in the non-parametric bootstrap procedure for obtaining confidence intervals of estimates based on the k largest values in a sample, the computations can be carried out in a more computationally efficient way by drawing bootstrap samples from the K0 (K0>k) largest values of the sample rather than from the entire sample. They propose that K0 be fixed at a value leading to a very low probability of drawing fewer than the required k largest entries of the sample and provide the expression of that probability.

The article is concise and well-written. The suggested approach appears to be useful for applications such as those considered in examples 1 and 2 (empirical percentile).

However, I have doubts about the correctness of the non-parametric bootstrap procedure for obtaining confidence intervals of GPD return value estimates as described in Example 3.

I have two major comments that I would like the authors to address or at least consider that they, despite not being covered by the article, should also be taken into consideration when bootstrapping to obtain confidence intervals estimates related to extremes.

Major comments:

1. I was not aware of the idea of the bootstrap being applied to the entire dataset rather than to a sample of cluster peaks as in the computation of confidence intervals of Example 3. In the usual form of the parametric bootstrap one does not return to the entire sample, but considers the (much smaller) sample to which the GPD was fitted. In any case, ensuring that the coverage rates - the percentage of times that a confidence interval really contains the true parameter in (hypothetical) repetitions of the same sampling and estimation process - of bootstrap confidence intervals are sufficiently correct has, in my view, priority over the computational efficiency of those intervals. Both Coles and Simiu (2003, J. Engrg. Mech., 129 (11), 1288-1294) and Schendel and Thongwichian (2017, Adv. Water Resour., 99, 53-59, http://dx.doi.org/10.1016/j.advwatres.2016.11.011) consider the shortcomings of bootstrap intervals with respect to coverage, the first paper offering an ad hoc solution and the second suggesting the use of Test Inversion Bootstrap. I wonder if the authors could add information to the article about the coverage rates of their confidence intervals.

2. The results shown in figures 3, 4 and 5 are based on M=10,000 bootstrap replications, while those shown in Figure 8 are based on M=1,000. I wonder if the authors could say something about how M should be chosen. According to Efron and Tibshirani (1993, Monographs on Statistics & Applied Probability 57), 200 bootstrap replications are usually enough for obtaining reasonable estimates of the standard error. Could optimizing the number of bootstrap replications be a possible solution to some of the

computational problems pointed out by the authors?

Specific comments:

 c Page 1, Line 3: 'confidence intervals . . . can be estimated'. I would replace 'estimated' with 'obtained' everywhere, since the intervals are random variables and not parameters.

 c Page 1, Line 13: In the light of my Major Comment 1, I would not say that "This is a straightforward procedure"; it is not the computational or algorithmic aspects of a method that matter most, but its validity.

---

## Author Comment (AC1) · 17 Jan 2017

Response to Reviewer 1, R Katz:

*Thank you for your thorough review. Our responses are in italics below. Please note that there is considerable overlap with the comments from the other reviewer, and we refer to our separate response to her). We propose to include a new Fig 6 which looks at the CIs as a function of number of bootstrap resamples. We will also include a new paragraph to the Discussion. This is outlined in our reply to Reviewer 2.*

How to Bootstrap Extremes if You Must GENERAL COMMENTS: The focus of the manuscript is on efficient use of the bootstrap, a resampling tech- nique, to quantify uncertainty (e.g., in the form of a confidence interval) in estimated extreme statistics such as return levels. Justification is provided for a simplified boot- strap procedure

in which the resamples are generated through only drawing from the highest values in the original sample, not the entire sample. This common sense result is consistent with conventional statistical modeling of extremes, with the common as- sumption that the uncertainty in estimating the rate of exceedance of a high threshold can be ignored (e.g., Chapter 4 in Coles, 2001). Perhaps the present paper serves to place this conventional approach on firmer footing.

Nevertheless, there are a number of alternative techniques for uncertainty quantification in extreme value analysis not even mentioned in the manuscript. These alternatives include different implementations of the bootstrap, as well as ones in which no resampling need be performed (e.g., profile likelihood technique; Coles, 2001). At the least, these alternatives should be mentioned.

*We agree, and we will include a section where we go through the various alternatives to non-parametric bootstrapping. However, we will argue that this is somewhat beside the point of the article as our main objective has been to investigate how tail statistics can be bootstrapped, if, as the referee says, you must. We do not necessarily argue that non-parametric bootstrapping is the best alternative, and we will make clearer where we think it is appropriate to use (see also our reply to Reviewer 2).*

For this reason, I recommend that the manuscript be accepted for publication subject to minor revision. SPECIFIC COMMENTS: (1) Nonparametric versus parametric bootstrap A nonparametric bootstrap is used in which the resamples are created by drawing with replacement from the original sample. When fitting extreme value distributions (e.g., the generalized Pareto in Sec. 3.3), it has been suggested that a parametric boot- strap would be preferable for constructing confidence intervals for return levels (i.e., resamples are created by Monte Carlo simulation from the fitted distribution) (Kysely, 2008).

*We agree that, especially for small samples, parametric bootstraps are probably capable of better coverage than non-parametric bootstrap techniques. However, we are*

*looking at very large samples, indeed at samples that are so large that we can perform in some cases in-sample estimates of 100-year return values. The paper now acknowledges the limitations of non-parametric bootstraps, and we stress that it should be seen as a study of how to efficiently handle the original data set if, as the reviewers points out, you wish to perform a non-parametric bootstrap. We will include a paragraph in the Discussion where we look at the caveats to using non-parametric bootstraps (see our reply to Reviewer 2).*

(2) Refined bootstrap techniques Bootstrap-based confidence intervals can be too short, especially for return levels with long return periods. Consequently, alternative more involved bootstrap techniques (e.g., the so-called "test inversion" bootstrap) have been proposed to improve the per- formance of such confidence intervals (Schendel and Thongwichian, 2015).

*This is an interesting technique, and Reviewer 2 also refers to a follow-up paper by the same authors. We will include a short paragraph in the Discussion where we outline this alternative method (see our reply to Reviewer 2).*

(3) Alternatives to bootstrap When estimating the parameters of an extreme value distribution by maximum likeli- hood, an alternative technique for obtaining confidence intervals for return levels is pro- file likelihood (Coles, 2001). This technique does not require any resampling, but does require repeated fits of the extreme value distribution under parameter constraints. It is competitive with resampling for obtaining confidence intervals of return levels (e.g., Schendel and Thongwichian, 2015).

*The profile likelihood technique is a well-known technique, but it falls outside the scope of this paper to investigate it as we focus strictly on efficient methods for non-parametric bootstrapping.*

REFERENCES: Kysely, J.: A cautionary note on the use of nonparametric bootstrap for estimating un- certainties in extreme-value models, Journal of Applied Meteorology and Climatology, 47, 3236–3251, 2008. Schendel, T. and Thongwichian, R.: Flood

frequency analysis: Confidence interval es- timation by test inversion bootstrapping, Advances in Water Resources, 83, 1–9, 2015.

---

## Author Comment (AC2) · 17 Jan 2017

Response to Reviewer 2, Sofia Caires:

*Thank you for your thorough review. Our responses are in italics below. Please note that there is considerable overlap with the comments from the other reviewer, and we refer to our separate response to him). We propose to include a new Fig 6 (attached) which looks at the CIs as a function of number of bootstrap resamples. We will also include a new paragraph to the Discussion. See below for details.*

General comments: The authors show that in the non-parametric bootstrap procedure for obtaining confidence intervals of estimates based on the k largest values in a sample, the computations can be carried out in a more computationally efficient way by drawing bootstrap samples from the K0 (K0>k) largest values of the sample rather

[Figure]

than from the entire sample. They propose that K0 be fixed at a value leading to a very low probability of drawing fewer than the required k largest entries of the sample and provide the expression of that probability.

The article is concise and well-written. The suggested approach appears to be useful for applications such as those considered in examples 1 and 2 (empirical percentile). However, I have doubts about the correctness of the non-parametric bootstrap procedure for obtaining confidence intervals of GPD return value estimates as described in Example 3.

I have two major comments that I would like the authors to address or at least consider that they, despite not being covered by the article, should also be taken into consideration when bootstrapping to obtain confidence intervals estimates related to extremes.

Major comments: 1. I was not aware of the idea of the bootstrap being applied to the entire dataset rather than to a sample of cluster peaks as in the computation of confidence intervals of Example 3. In the usual form of the parametric bootstrap one does not return to the entire sample, but considers the (much smaller) sample to which the GPD was fitted. In any case, ensuring that the coverage rates - the percentage of times that a confidence interval really contains the true parameter in (hypothetical) repetitions of the same sampling and estimation process - of bootstrap confidence intervals are sufficiently correct has, in my view, priority over the computational efficiency of those intervals. Both Coles and Simiu (2003, J. Engrg. Mech., 129 (11), 1288-1294) and Schendel and Thongwichian (2017, Adv. Water Resour., 99, 53-59, http://dx.doi.org/10.1016/j.advwatres.2016.11.011) consider the shortcomings of bootstrap intervals with respect to coverage, the first paper offering an ad hoc solution and the second suggesting the use of Test Inversion Bootstrap. I wonder if the authors could add information to the article about the coverage rates of their confidence intervals.

*The reason we return to the entire sample in Example 3 is that the data set represents independent forecasts (taken at long lead times, as described by Breivik et al, 2013,*

[Figure]

*2014). We are thus in the situation where we are not limited to a peaks-over-threshold technique but can (and should) resample from the entire sample and then set a threshold (note the difference between a POT and a threshold). It was the magnitude of this data set that motivated us to explore which simplifications can be made in order to speed up the bootstrapping for tail statistics. We will elaborate on this in our revision of Example 3 to make clearer why it is important to revisit the entire sample.*

*As for the question of whether a non-parametric bootstrapping method will underestimate the width (coverage) of CIs, we agree in general, but note that our examples involve very large data sets. See also our reply to Reviewer 1.*

2. The results shown in figures 3, 4 and 5 are based on M=10,000 bootstrap replications, while those shown in Figure 8 are based on M=1,000. I wonder if the authors could say something about how M should be chosen. According to Efron and Tibshirani (1993, Monographs on Statistics  Applied Probability 57), 200 bootstrap replications are usually enough for obtaining reasonable estimates of the standard error. Could optimizing the number of bootstrap replications be a possible solution to some of the computational problems pointed out by the authors?

*Although it is certainly true that M=10,000 bootstrap replications is excessive, 200 may in some cases be on the low side. We found in our global study of return values for marine wind and significant wave height (Breivik et al, 2014, supplementary figure 7) that the confidence intervals tend to stabilize around 500 bootstrap replications when we look at GPD return estimates. We have chosen a very high number of bootstrap replications here for no better reason than because we could afford it, and because for some tail parameters it is desirable. We will include a figure which shows the convergence of the CIs as a function of the number of bootstrap resamples from 50 to 10,000 for non-parametric in-sample estimates of the 100-year return value for significant wave height (see below). The figure shows that indeed for the data set considered we can settle for 1,000 or perhaps slightly fewer bootstrap replications, but probably not as little as 200. To go with the figure below we will include the following text:*

*It is also of interest to investigate just how many bootstrap resamples are actually needed to obtain CIs from a non-parametric bootstrap technique. In Fig 5 we chose M = 10,000. As Fig 6 [new] shows, this is clearly excessive for reasonable thresholds K0. In fact, Efron and Tibshirani (1993) state that 200 resamples are normally enough. We find this to be on the low side in our case, as Fig 6 shows. However, 1000 resamples is sufficient in this case, but this should be investigated in each case. Breivik et al (2014) found (see their Supplementary Fig 7) that for a similar data set, 500 resamples would be sufficient when employing a GPD technique.*

Specific comments: Page 1, Line 3: "confidence intervals . . . can be estimated". I would replace "estimated" with "obtained" everywhere, since the intervals are random variables and not parameters.

*Agreed.*

Page 1, Line 13: In the light of my Major Comment 1, I would not say that "This is a straightforward procedure"; it is not the computational or algorithmic aspects of a method that matter most, but its validity.

*We agree, and we plan to make the appropriate changes to the manuscript by emphasizing that the procedure is straightforward, but the method of non-parametric bootstrapping has been found to lead to too narrow CIs (low coverage). We suggest to incorporate the following text in the discussion:*

*We have investigated the conditions that must be met to form a non-parametric bootstrap for tail statistics such as return values (which depend on all three parameters of the GEV or GPD). An important question is whether non-parametric bootstraps yields CIs with sufficient coverage, ie, CIs that are wide enough. This has been extensively studied by Kysely (2008) who found that non-parametric bootstraps in particular, but also parametric bootstraps tend to have too low coverage. This problem is not addressed by our study, and it is clear that alternative methods are often called for. In particular, the Test Inversion Bootstrap advocated by Schendel and Thongwichian (2015,*

*2017) is a promising method. However, non-parametric tail statistics are often a necessary first approach to obtaining CIs, and the results presented show that we can comfortably assume that the results will remain unchanged if we take a small subset of the original data set, provided we follow the procedure outlined in Section 2.*

[Figure]

**In-sample 100-year return level estimates**

Significant wave height [m]

Number of bootstrap resamples, $M$

Mean
Std dev

**Fig. 1.** New Fig 6: Mean and standard dev on 100-yr in-sample return estimates with a threshold K0 = 1,000 as a function of number of bootstrap resamples, M. For M > 1000 the CIs are quite stable.

---

## Author Response (AR1)

**Response to Reviewer 1, R Katz**

*Thank you for your thorough review. Our responses are in red italics below. Please note that there is considerable overlap with the comments from the other reviewer, and we refer to our separate response to her). We propose to include a new Fig 6 which looks at the CIs as a function of number of bootstrap resamples. We will also include a new paragraph to the Discussion. This is outlined in our reply to Reviewer 2.*

GENERAL COMMENTS:

The focus of the manuscript is on efficient use of the bootstrap, a resampling technique, to quantify uncertainty (e.g., in the form of a confidence interval) in estimated extreme statistics such as return levels. Justification is provided for a simplified bootstrap procedure in which the resamples are generated through only drawing from the highest values in the original sample, not the entire sample. This common sense result is consistent with conventional statistical modeling of extremes, with the common assumption that the uncertainty in estimating the rate of exceedance of a high threshold can be ignored (e.g., Chapter 4 in Coles, 2001). Perhaps the present paper serves to place this conventional approach on firmer footing. Nevertheless, there are a number of alternative techniques for uncertainty quantification in extreme value analysis not even mentioned in the manuscript. These alternatives include different implementations of the bootstrap, as well as ones in which no resampling need be performed (e.g., profile likelihood technique; Coles, 2001). At the least, these alternatives should be mentioned.

*We agree, and we now mention the weaknesses of non-parametric bootstrapping with reference to Kysely (2008) on page 1, line 13. We also include a brief discussion of the test inversion bootstrap method in the Discussion (page 7, lines 5-13). However, we maintain that this is somewhat beside the point of the article as our main objective has been to investigate how tail statistics can be bootstrapped, if, as the referee says, you must. We do not necessarily argue that non-parametric bootstrapping is the best alternative, and we have made clearer where we think it is appropriate to use (see also our reply to Reviewer 2).*

For this reason, I recommend that the manuscript be accepted for publication subject to minor revision.

SPECIFIC COMMENTS: (1) Nonparametric versus parametric bootstrap. A nonparametric bootstrap is used in which the resamples are created by drawing with replacement from the original sample. When fitting extreme value distributions (e.g., the generalized Pareto in Sec. 3.3), it has been suggested that a parametric bootstrap would be preferable for constructing confidence intervals for return levels (i.e., resamples are created by Monte Carlo simulation from the fitted distribution) (Kysely, 2008).

*We agree that, especially for small samples, parametric bootstraps are probably capable of better coverage than non-parametric bootstrap techniques. However, we are looking at very large samples, indeed at samples that are so large that we can perform in some cases in-sample estimates of 100-year return values. The paper now acknowledges the limitations of non-parametric bootstraps, and we stress that it should be seen as a study of how to efficiently handle the original data set if, as the reviewers points out, you wish to perform a non-parametric bootstrap. We have included a paragraph in the Discussion (page 7, lines 5-13) where we look at the caveats to using non-parametric bootstraps (see also our reply to Reviewer 2).*

(2) Refined bootstrap techniques Bootstrap-based confidence intervals can be too short, especially for return levels with long return periods. Consequently, alternative more involved bootstrap techniques (e.g., the so-called "test inversion" bootstrap) have been proposed to improve the performance of such confidence intervals (Schendel and Thongwichian, 2015).

*This is an interesting technique, and Reviewer 2 also refers to a follow-up paper by the same authors. We have included a short paragraph in the Discussion (page 7, lines 5-13) where we outline this alternative method (see also our reply to Reviewer 2).*

(3) Alternatives to bootstrap When estimating the parameters of an extreme value distribution by maximum likelihood, an alternative technique for obtaining confidence intervals for return levels is profile likelihood (Coles, 2001). This technique does not require any resampling, but does require repeated fits of the extreme value distribution under parameter constraints. It is competitive with resampling for obtaining confidence intervals of return levels (e.g., Schendel and Thongwichian, 2015).

*The profile likelihood technique is a well-known technique, but it falls outside the scope of this paper to investigate it as we focus strictly on efficient methods for non-parametric bootstrapping.*

**Response to Reviewer 2, S Caires**

*Thank you for your thorough review. Our responses are in red italics below. Please note that there is considerable overlap with the comments from the other reviewer, and we refer to our separate response to him). We propose to include a new Fig 6 (attached) which looks at the CIs as a function of number of bootstrap resamples. We will also include a new paragraph to the Discussion. See below for details.*

5    General comments:

The authors show that in the non-parametric bootstrap procedure for obtaining confidence intervals of estimates based on the k largest values in a sample, the computations can be carried out in a more computationally efficient way by drawing bootstrap samples from the K0 (K0>k) largest values of the sample rather than from the entire sample. They propose that K0 be fixed

10   at a value leading to a very low probability of drawing fewer than the required k largest entries of the sample and provide the expression of that probability. The article is concise and well-written. The suggested approach appears to be useful for applications such as those considered in examples 1 and 2 (empirical percentile). However, I have doubts about the correctness of the non-parametric bootstrap procedure for obtaining confidence intervals of GPD return value estimates as described in Example 3. I have two major comments that I would like the authors to address or at least consider that they, despite not being

15   covered by the article, should also be taken into consideration when bootstrapping to obtain confidence intervals estimates related to extremes.

Major comments:

1. I was not aware of the idea of the bootstrap being applied to the entire dataset rather than to a sample of cluster peaks as in the computation of confidence intervals of Example 3. In the usual form of the parametric bootstrap one does not return to the

20   entire sample, but considers the (much smaller) sample to which the GPD was fitted. In any case, ensuring that the coverage rates - the percentage of times that a confidence interval really contains the true parameter in (hypothetical) repetitions of the same sampling and estimation process - of bootstrap confidence intervals are sufficiently correct has, in my view, priority over the computational effi- ciency of those intervals. Both Coles and Simiu (2003, J. Engrg. Mech., 129 (11), 1288-1294) and Schendel and Thongwichian (2017, Adv. Water Resour., 99, 53-59, http://dx.doi.org/10.1016/j.advwatres.2016.11.011)

25   consider the shortcomings of bootstrap intervals with respect to coverage, the first paper offering an ad hoc solution and the second suggesting the use of Test Inversion Bootstrap. I wonder if the authors could add information to the article about the coverage rates of their confidence intervals.

*The reason we return to the entire sample in Example 3 is that the data set represents independent forecasts (taken at long lead times, as described by Breivik et al, 2013, 2014). We are thus in the situation where we are not limited to a peaks-over-*

30   *threshold technique but can (and should) resample from the entire sample and then set a threshold (note the difference between a POT and a threshold). It was the magnitude of this data set that motivated us to explore which simplifications can be made in order to speed up the bootstrapping for tail statistics. We have elaborated on this in our revision of Example 3 (see p 5, lines 25-27) to make clearer why it is important to revisit the entire sample. As for the question of whether a non-parametric bootstrapping method will underestimate the width (coverage) of CIs, we agree in general, but note that our examples involve*

35   *very large data sets. See also our reply to Reviewer 1.*

2. The results shown in Figs 3, 4 and 5 are based on M=10,000 bootstrap replications, while those shown in Fig 8 are based on M=1,000. I wonder if the authors could say something about how M should be chosen. According to Efron and Tibshirani (1993, Monographs on Statistics Applied Probability 57), 200 bootstrap replications are usually enough for obtaining reasonable estimates of the standard error. Could optimizing the number of bootstrap replications be a possible solution to some

40   of the computational problems pointed out by the authors?

*Although it is certainly true that M=10,000 bootstrap replications is excessive, 200 may in some cases be on the low side. We found in our global study of return values for marine wind and significant wave height (Breivik et al, 2014, supplementary figure 7) that the confidence intervals tend to stabilize around 500 bootstrap replications when we look at GPD return estimates. We have chosen a very high number of bootstrap replications here for no better reason than because we could afford it, and*

45   *because for some tail parameters it is desirable. We have included a new Fig 6 which shows the convergence of the CIs as a function of the number of bootstrap resamples from 50 to 10,000 for non-parametric in-sample estimates of the 100-year return value for significant wave height. The figure shows that indeed for the data set considered we can settle for 1,000 or perhaps slightly fewer bootstrap replications, but probably not as little as 200. To go with Fig 6 we include the following text (page 5,*

*lines 4-9):*

*"It is also of interest to investigate just how many bootstrap resamples are actually needed to obtain CIs from a non-parametric bootstrap technique. In Fig 5 we chose M = 10,000. As Fig 6 shows, this is clearly excessive for reasonable thresholds $K_0$. In fact, Efron and Tibshirani (1993) state that 200 resamples are normally enough.We find this to be on the low side in our case, as Fig 6 shows. However, 1000 resamples is sufficient in this case, but this should be investigated in each case. Breivik et al. (2014) found (see their Supplementary Fig 7) that for a similar data set, 500 resamples would be sufficient when employing a Generalized Pareto Distribution (GPD) on threshold exceedances."*

Specific comments:

Page 1, Line 3: "confidence intervals . . . can be estimated". I would replace "estimated" with "obtained" everywhere, since the intervals are random variables and not parameters.

*Agreed.*

Page 1, Line 13: In the light of my Major Comment 1, I would not say that ?This is a straightforward procedure?; it is not the computational or algorithmic aspects of a method that matter most, but its validity.

*We agree, and we have rewritten the Introduction to emphasize that the procedure is straightforward, but the method of non-parametric bootstrapping has been found to lead to too narrow CIs (low coverage), see page 1, line 13-14. We have also included the following text in the Discussion (page 7, lines 5-15): "As mentioned in the Introduction, an important question is whether non-parametric bootstraps yield CIs with sufficient coverage, ie, CIs that are wide enough. This has been extensively studied by Kysely (2008) who found that non-parametric bootstraps in particular, but also parametric bootstraps tend to have too low coverage. This problem is not addressed by our study, and it is clear that alternative methods are often called for. In particular, the Test Inversion Bootstrap (Carpenter, 1999) is a promising method where the test inversion refers to the duality between hypothesis 10 testing and confidence intervals. Schendel and Thongwichian (2015, 2017) show how this method, originally developed for estimation of statistics of single parameters in the presence of nuisance parameters, can be extended to handle return levels which depend on three parameters for both the Generalized Extreme Value Distribution and GPD by utilizing a maximum likelihood technique. However, non-parametric bootstraps represent a quick and hypothesis-free approach to obtaining CIs, and as the results presented show we can comfortably assume that the results will remain unchanged if we select a small subset of the original sample, provided we follow the procedure outlined in Section 2.??*

*Below is a full account of the differences between this version and the previous version of the manuscript.*

Yours sincerely,

Øyvind Breivik and Ole Johan Aarnes,
Bergen, 2017-02-21

[revised manuscript text omitted]